# Interfacial Properties of Pea Protein Hydrolysate: The Effect of Ionic Strength

Krystalia Sarigiannidou [1,†], Davide Odelli [1,2,†], Flemming Jessen [1], Mohammad Amin Mohammadifar [1], Fatemeh Ajalloueian [3], Mar Vall-llosera [1,4], Antonio Fernandes de Carvalho [2] and Federico Casanova [1,*]

1 Research Group for Food Production Engineering, National Food Institute, Technical University of Denmark, Søltofts Plads, 2800 Kongens Lyngby, Denmark
2 Departamento de Tecnologia de Alimentos, Universidade Federal de Viçosa (UFV), Viçosa 36570-900, Brazil
3 Center for Intelligent Drug Delivery and Sensing Using Microcontainers and Nanomechanics (IDUN), Department of Health Technology, Technical University of Denmark, 2800 Kongens Lyngby, Denmark
4 Department of Biology and Biological Engineering, Food and Nutrition Science, Chalmers University of Technology, SE-41296 Gothenburg, Sweden
* Correspondence: feca@food.dtu.dk; Tel.: +45-4525-2716
† These authors contributed equally to this work.

**Abstract:** The effect of a tryptic hydrolysis as well as the effect of ionic strength (0–0.4 M NaCl) was investigated on the oil/water interfacial properties of soluble pea protein hydrolysate (SPPH) at neutral pH and room temperature (20 ± 0.01 °C). SEC-MALS and SDS-Page analysis showed that tryptic hydrolysis created a lower molecular weight polypeptide mixture, whereas FTIR analysis and DSC thermograms demonstrated a more disordered and flexible structure. The bulk properties of SPPH were studied in terms of hydrodynamic diameter and turbidity, where higher particle size (+ ~13 nm) and turbidity were observed at 0.4 M NaCl. Regarding the interfacial properties, the surface activity of SPPH improved by increasing ionic strength, with maximum interfacial pressure (14.28 mN/m) at 0.4 M NaCl. Nevertheless, the addition of NaCl negatively affected the elasticity and strength of the interfacial film, where the sample without salt exhibited the highest dilatational and shear storage modulus in all the frequencies considered.

**Keywords:** pea protein; enzymatic hydrolysis; interfacial properties; ionic strength

## 1. Introduction

Proteins are amphiphilic macromolecules consisting of a sequence of amino acids linked by polypeptide bonds. The polypeptide chains are stabilized by hydrogen bonds, electrostatic interactions, van der Waals interactions and hydrophobic interactions. They are linked by disulfide bonds and rearranged until they form the quaternary structure of the protein [1]. Owing to their amphiphilic nature, proteins can adsorb on a thermodynamically unstable interface, where they unfold and create an interfacial film [2]. Physicochemical properties of the protein, such as size, net charge and amino acid composition [1] and also parameters, such as pH and ionic strength, significantly influence the interfacial and emulsifying properties [3].

Increasing interest in plant-based proteins was observed in the previous years, as they are considered more sustainable ingredients with fewer diet limitations. The emulsifying ability of plant-derived proteins, such as pea, has been already investigated [4]. However, their applications are limited due to poor solubility and an unpleasant taste compared to whey protein [5,6].

To overcome these drawbacks, modification techniques, such as enzymatic hydrolysis [7–9], have been developed. The employment of enzymatic hydrolysis results in the formation of a lower molecular weight peptide mixture, with less quaternary and tertiary

structure, compared to the native protein [7]. Consequently, the diffusion rate and the affinity of the protein to the interface are increased [10,11].

The implementation of enzymatic hydrolysis on pea protein with different enzymes was investigated by García Arteaga et al. [12]. The authors showed that this method can improve the functional properties of the protein, while the trypsin hydrolysate exhibited the highest emulsifying (719 mL/g) capacity. In another study, Klost and Drusch [8] examined the effect of partial tryptic hydrolysis on the emulsifying and interfacial properties of pea protein at pH varying from 3 to 7. They found that this modification technique improved both the solubility and the emulsifying ability of the protein and influences the interfacial properties.

Parameters such as protein concentration, pH value, ionic strength and heat treatment significantly affect the protein structure and therefore influence the emulsion stability [2]. Tan et al. [13] studied the influence of ionic strength on the interfacial properties of the walnut protein-xanthan gum complex. They found maximum values of interfacial pressure at 0.35 M of NaCl and a higher dilatational modulus at 0.15 M of NaCl. Tian et al. [3] observed improvement in β-conglycinin adsorption in an oil/water interface with the addition of NaCl, yet the viscoelasticity of the interfacial film decreased.

This study aims to investigate the changes in the structure and size of pea protein after tryptic hydrolysis and the effect of ionic strength on oil/water interfacial properties of the hydrolysates. Indeed, a deeper knowledge of protein and peptides behavior at the oil/water interface would allow for the development of food colloids and surfactants based on natural ingredients and characterized by the desired features to manufacture emulsions, such as beverages and sauces, with greater stability and functionality.

## 2. Materials and Methods

### 2.1. Materials

Pea protein (PPI) Nutralys F85 F (80% protein content) was kindly provided by Roquette Frères (Lestrem, France). Trypsin from porcine pancreas (1000–2000 BAEE units/mg solid) was used as an enzyme and NaCl as salt, both purchased by Sigma (St. Louis, MO, USA). For the interfacial experiments was used sunflower oil (Ollineo, Netto, Denmark). The protein content of the soluble pea protein hydrolysate (SPPH) was 40%, measured by the DUMAS method. All concentrations were referred to protein content and all solutions were prepared in phosphate buffer (50 mM $Na_2HPO_4$, $NaH_2PO_4$, 150 mM NaCl, pH 7.2) unless otherwise stated.

### 2.2. Enzymatic Hydrolysis

Enzymatic hydrolysis was performed as described by Tamm et al. [7] with some modifications. Brief, 2% PPI suspension was prepared and stirred overnight at room temperature (~20 °C). Then, before enzyme addition, the protein suspension was transferred to a 500 mL Erlenmeyer flask, heated at 50 °C and pH adjusted to 8 using an aliquot amount of NaOH 0.5 M. The enzyme-substrate ratio was 1:50. The degree of hydrolysis was estimated with a pH-stat method as stated in (Equation (1)).

$$DH = \frac{h}{htot} * 100\% \qquad (1)$$

where *h* are the equivalents of peptide bonds hydrolyzed per g of protein and *htot* is the total peptide bonds expressed as equivalents of peptide bonds per g of protein. In that case, *htot* value was 7.2 meqv/g.

The treatment time was based on the *DH* value and lasted approximately 10 min. When a *DH* of 4% was reached, the Erlenmeyer flask was placed immediately in a water-bath at 75 °C for 30 min for the enzyme inactivation, followed by a cooling process until room temperature (~20 °C). Afterward, the protein suspension was centrifuged (12,000× *g*, 10 min, 20 °C) and the soluble pea protein hydrolysate (SPPH) was collected and lyophilized.

### 2.3. Size-Exclusion Chromatography (SEC-MALS)

The molecular weight of PPI and SPPH was determined using size-exclusion chromatography. For that purpose, 2 mg/mL sample was prepared in phosphate buffer (pH 7.2) and filtered with 0.1 μm pore size filter. The HPLC (Agilent, Santa Clara, CA, USA) was equipped with WTC-015S5 column (300 × 7.8 mm, 150 Å maximum pore size) Wyatt Technology, Santa Barbara, CA, USA). The elute was monitored in succession by a UV detector at 280 nm, a DAWN 8 light-scattering detector (Wyatt Technology, Santa Barbara, CA, USA) and an Optilab differential refractometer (Wyatt Technology). The flow rate was 0.8 mL/min and the injection volume 50 μL. The mobile phase was phosphate buffer, pH 7.2, containing 200 μL/L proClin (Sigma, St. Louis, MO, USA) to mitigate microbial growth. The buffer was prior filtered with a sterile single-use vacuum filter (Thermo Fisher Scientific, Roskilde, Denmark) with a pore size of 0.1 μm. Specific-refractive-index increments (dn/dc) of sample solution was 0.185 mL/g. Data analysis and molecular weight calculations were performed using the ASTRA software (Wyatt technology, Dernbach, Germany, Version 7.3.2).

### 2.4. Electrophoretic Study (SDS-PAGE)

SDS-PAGE analysis was performed as described by Laemmli [14] using 12% acrylamide (C = 2.6% ($w/w$)) slab gels (1.5 mm thick). Brief, 2 mL of 1% sodium dodecyl sulfate (SDS) in 100 mM dithiothreitol (DTT) and 60 mM Tris HCl (pH 8.3) were added in 50 mg of dry protein and then the sample was slightly shaken for 1 h. Afterward, the sample was homogenized (Polytron PT 1200, Kinematica) for 30 s, boiled for 2 min and incubated at room temperature for 30 min. This process was performed twice followed by centrifugation at 20,000× $g$ for 15 min at 20 °C. The supernatant was collected and diluted in a buffer with 125 mM tris HCl (pH 6.8), 2.4% SDS, 50 mM DTT, 10% glycerol, 0.5 mM EDTA and bromophenol blue. The gel was loaded with 10 μL of the diluted sample, corresponding to 40 μg of protein, considering the protein content of the dry sample. The electrophoresis was carried out at a constant voltage of 100 V for 15 min followed by 150 V for 1 h. Finally, the gel was stained with colloidal Coomassie Brilliant Blue. As markers were used Mark12™ from Novex.

### 2.5. Fourier Transform Infrared Spectroscopy (FTIR)

PPI and SPPH were subjected to FTIR spectrum analysis, as described by Casanova et al. [15], with a PerkinElmer Spectrum 100 FT-IR spectrometer (Waltham, MA, USA). The samples were analyzed in a wavelength from 650 to 4000 cm$^{-1}$, with a resolution of 4 cm$^{-1}$ and the signal was recorded in 4 scans.

### 2.6. Differential Scanning Calorimetry (DSC)

The thermal properties of PPI and SPPH were studied by differential scanning calorimetry DSC 250 (TA Instruments, New Castle, DE, USA), equipped with Refrigerated Cooling System 90. Distilled water was used for calibration of the instrument for heat flow and temperature (melting point (m.p.) = 0 °C; DHm = 334 J/g) and indium (m.p. = 156.5 °C; DHm = 28.5 J/g) and nitrogen (50 mL/min) as carrier gas. Approximately 5 mg of dry samples were placed in aluminum pans of 30 μL and sealed with a hermetic lid. An empty pan was used as a reference. The samples were heated from 0 °C to 250 °C at a heating rate of 5 °C/min. Thermograms were analyzed from their total, reversible and non-reversible heat flow. Glass transition ($T_g$) was identified by the reversible heat flow line, whereas denaturation temperature ($T_d$) and melting point ($T_m$) by the total heat flow line. The enthalpy was determined by the area of the endothermic peak. All the thermograms were analyzed using Trios software. All the samples were measured in triplicate.

### 2.7. Hydrodynamic Diameter

Hydrodynamic diameter refers to how a particle diffuses within a fluid and can thus be defined as the diameter of a sphere having the same diffusion coefficient of the particle.

The measurement of the hydrodynamic diameter of the protein solution was conducted by Dynamic Light Scattering (DLS) by using Zetasizer Nano-ZS (Malvern Instruments, Worcestershire, UK) equipped with capillary cells. For this analysis, 1% protein solution was prepared in milli-Q water, afterwards, NaCl was added in different concentrations (0.1, 0.2 and 0.4 M) and the solutions were stirred for 30 min. Finally, the samples were diluted 1:100 with milli-Q water and let rest for 5 min before the analysis. The samples presented a viscosity of $10^{-3}$ Pa.s and their analysis was carried out at room temperature (24 ± 1 °C), with a scattering angle of 173° and a wavelength of 633 nm. Hydrodynamic diameter, *Dh*, was calculated as stated by the Stokes–Einstein equation (Equation (2)),

$$Dh = \frac{KB * T}{3 * \pi * \eta * Dt} \tag{2}$$

where *KB* is the Boltzmann's constant, *T* is the temperature, $\pi$ is the mathematical constant pi, $\eta$ the solvent viscosity (Pa s$^{-1}$) and *Dt* is the diffusion coefficient extracted from the fit of the correlation curve using the cumulative method [15]. All the samples were measured in triplicate and considered relatable if a polydispersity index (PdI) less than 0.5 was obtained for each measurement. All the measurements here reported, presented PdI oscillating between 0.1 and 0.3.

*2.8. Turbidity*

The turbidity measurements were carried out with a U-I500 spectrophotometer (HI-TACHI, Tokyo, Japan) at 400 nm. For this analysis, 1% SPPH was prepared in milli-Q water and stirred for 1 h, followed by NaCl addition in different concentrations and stirring for 30 min. Solutions were diluted 1:10 before analysis. Milli-Q water was used as blank. All the samples were measured in triplicate.

*2.9. Interfacial Pressure and Interfacial Dilatational Properties*

Interfacial pressure and interfacial dilatational properties of SPPH in water/oil interface were measured by the pendant drop technique, using an optical drop tensiometer OCA 25 (DataPhysics Instruments, Filderstadt, Baden-Württemberg, Germany). The interfacial tension values of the samples were calculated based on the Young-Laplace equation, monitoring the shape of a droplet Equation (3). Where $\Delta p$ is the Laplace pressure, $\gamma$ is the surface tension, and *R*1 and *R*2 are the radii curvature of the formed droplet.

$$\Delta p = -\gamma \left( \frac{1}{R1} + \frac{1}{R2} \right) \tag{3}$$

$$\boldsymbol{\pi} = \gamma 0 - \gamma \tag{4}$$

For that purpose, 0.5% of SPPH was prepared in phosphate buffer (pH 7.2) and stirred for 1 h, followed by the addition of aliquot amount NaCl and 30 min stirring. Afterward, the solution was injected by a syringe into an optical glass cuvette filled with sunflower oil. The shape of a droplet with a volume of 35 μL was then monitored for 1800 s and the interfacial tension was measured every 2 s. Density of all the samples was measured 997 kg/m$^3$ while oil phase density was 919 kg/m$^3$. Distilled water was used as control; the interfacial tension was measured at 20.98 ± 0.21 mN/m. The interfacial pressure of all the solutions was then defined by Equation (4), where $\boldsymbol{\pi}$ is the interfacial pressure, $\gamma 0$ the interfacial tension of distilled water and $\gamma$ the interfacial tension of the protein solutions.

Adsorption kinetics parameters, $k_p$ and $k_r$, were determined by Equation (5).

$$\ln \left( \frac{\pi_e - \pi_\theta}{\pi_e - \pi_0} \right) = -k * \theta \tag{5}$$

where $\pi_0$, $\pi_\theta$ and $\pi_e$ are the interfacial pressure at the beginning, at any time point and equilibrium, respectively. *k* is the first-order rate constant and $\theta$ the time.

Interfacial dilatational properties were determined through an oscillatory test after the droplet was equilibrated for 1800 s following the described above protocol. Frequency sweep tests with a fixed amplitude of 2% were applied in a frequency range from 0.1 to 1 Hz. The tests were performed in 4 steps (0.1, 0.2, 0.4, 1 Hz), with 10 s waiting time between each step. The droplet was subjected to four cycles at each frequency. Dilatational storage (E′) and loss (E″) moduli and their loss factor tan $\delta$ (E″/E′) were obtained. All samples were measured in triplicate.

### 2.10. Shear Interfacial Rheology

Interfacial shear rheology measurements were conducted with a sensitive stress-controlled rheometer (DHR2, TA Instruments, New Castle, DE, USA) equipped with a double wall-ring (DWR) geometry. 20 mL of sample (10%) was poured into a Delrin® trough and an annular ring was placed at water/air interface. Subsequently, 20 mL of sunflower oil was added on the top. Amplitude sweep measurement was performed in strain range from 0.01 to 100% at a frequency of 0.2 Hz to determine the linear viscoelasticity region. Time sweep was carried out at a strain amplitude of 0.1% and frequency of 0.2 Hz for 25 h at constant temperature (20 $\pm$ 0.01 °C). The measurements were conducted in duplicate.

### 2.11. Statistical Analysis

The statistical analysis of hydrodynamic diameter, turbidity, interfacial pressure, and dilatational properties was performed with one-way ANOVA and pair comparison was achieved by Tukey's test at $p < 0.05$.

## 3. Results and Discussion

### 3.1. Molecular Weight Distribution

Molecular weight distribution results obtained by size exclusion chromatography (SEC-MALS) and electrophoretic analysis (SDS-PAGE) are summarized in Table 1.

**Table 1.** Molecular weight distribution of PPI and SPPH.

| SEC-MALS | | | | | | | |
|---|---|---|---|---|---|---|---|
| Molecular Weight (kDa) | | | | | | | |
| PPI | 920 | | 165 | | 68 | 50 | |
| SPPH | 60 | | 30 | | 10 | 9 | 2 |
| **SDS-PAGE** | | | | | | | |
| Molecular Weight (kDa) | | | | | | | |
| PPI | 90 | 66 | 50 | 40 | 30 | 21 | 18 | <6 |
| SPPH | 64 | 55 | 25 | 10 | <6 | | | |

SEC-MALS analysis (Figure 1) showed that PPI exhibited protein fractions of ~920, ~165, ~68 and ~50 kDa. Protein fractions with Mw higher than 920 kDa might be attributed to supramolecular legumin aggregates, as those reported by Yang et al. [16]. Similar protein fractions were also observed by Shevkani et al. [17] for field pea protein with purity over 90%. A fraction with Mw of ~165 kDa probably corresponds to vicilin; the molecular weight varies from 150 to 180 kDa [16,17]. A size of ~68 kDa fraction might represent legumin or/and convicilin subunits, as they have Mw of 60 and 70 kDa, respectively [9]. A lower Mw fraction (~50 kDa) was attributed to a mixture of legumin, vicilin, and convicilin subunits but also to albumins; the Mw varies from 6 to 80 kDa [16].

The hydrolysis of pea protein by trypsin caused a decrease in the Mw of all fractions. In particular, the major effects were observed for vicilin, where trypsin cleaved its quaternary structure into smaller peptides (Table 1). The obtained results are in agreement with the findings of Klost and Drusch [8] who reported that vicilin was cleaved even at lower *DH* of 2%. This might be caused due to the relatively high content of vicilin in arginine (6%) and lysine (6%) [18].

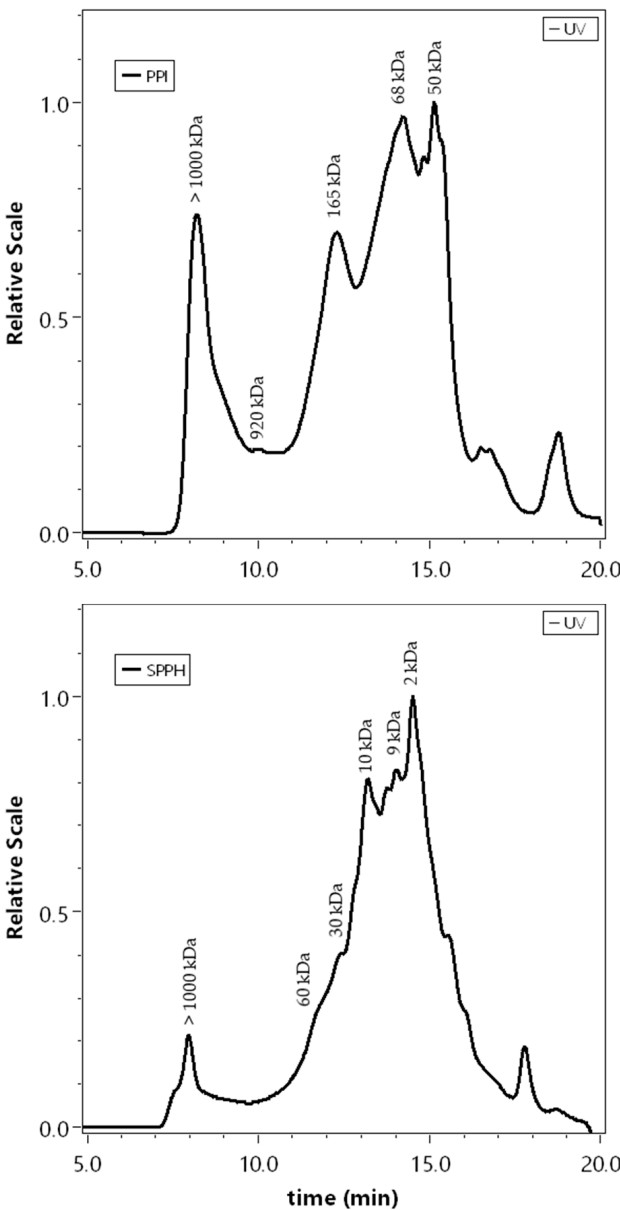

**Figure 1.** Size-exclusion chromatograms of PPI and SPPH.

The SDS-PAGE analysis of PPI (Figure 2) under reducing conditions showed main bands at ~90, ~66, ~50, ~40, ~30, ~21 and ~18 kDa. The subunit of ~90 kDa was attributed to lipoxygenase [19]. In several studies [9,17,19], subunits of ~50 and ~30 kDa had been ascribed to vicilin monomer, while lower Mw subunits (19–12.5 kDa) to vicilin fraction. In the same studies, fractions of ~40 and ~20 kDa corresponded to α-chain and β-chain of legumin, respectively.

In addition, the electrophoretic profile of SPPH (Table 1) confirmed a decrease in Mw of all fractions. In particular, bands of ~64, ~55, ~25, ~10 and lower than 6 kDa were observed. Similar results were reported by Klost et al. [9], where it was described that trypsin cleaved the α-chain of legumin, whereas the β-chain remained almost the same. According to the authors, this was attributed to the fact that legumin-α has a higher content of lysine and arginine compared to legumin-β.

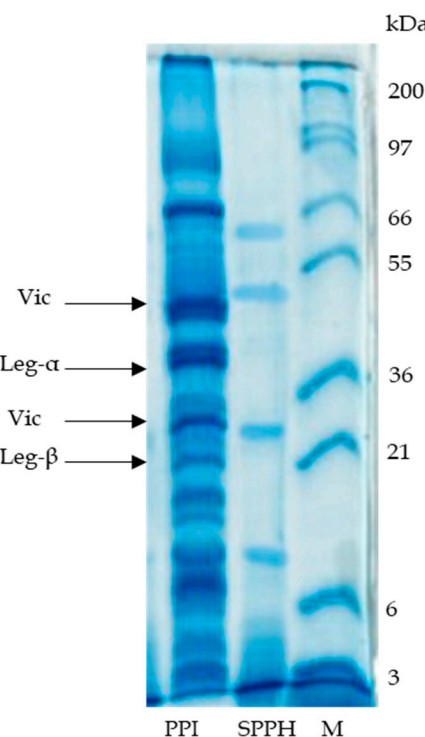

**Figure 2.** SDS-PAGE of PPI and SPPH. M is referred to marker.

*3.2. Fourier Transform Infrared Spectroscopy (FTIR)*

FTIR analysis is a non-destructive method that identifies the chemical composition of the whole sample and provides information about the functional groups and the protein structure [20].

In Figure 3, the first region was observed from 3000 to 3600 cm$^{-1}$, with maximum absorbance at 3275 cm$^{-1}$ for both PPI and SPPH. According to Acquah et al. [21], this region corresponds to Amide A and it is associated with the stretching vibrations of hydrogen bonds between O-H and N-H. PPI also exhibited a low-intensity peak at 3070 cm$^{-1}$, which is attributed to Amide B band. This band has been associated with the stretching vibrations of intramolecular hydrogen bonds between N-H groups [20]. The following region, from 3000 to 2500 cm$^{-1}$, is linked to C–H stretching and it mainly derives from lipids and carbohydrates [22].

Amide I and II regions that occur from 1600 to 1700 cm$^{-1}$ and 1500 to 1600 cm$^{-1}$, respectively, provide information regarding the secondary structure of proteins. Amide I is governed by the stretching vibrations of C = O (70–85%) and C-N groups and is linked to the backbone conformation [21]. No difference in wavenumber was observed between the two samples as both exhibited maximum absorbance at 1632 cm$^{-1}$, it can thus be considered that the secondary structure of the proteins involved may not be modified by the hydrolysis treatment. Amide II band derives primarily from the N-H bending and secondarily from the C-N stretching vibrations [23]. In that region, PPI exhibited a maximum of 1527 cm$^{-1}$, whereas SPPH exhibited a maximum of 1577 cm$^{-1}$. Amide III can be found in a wavenumber from 1400 to 1200 cm$^{-1}$ and it is a combination of N–H bending and C-N stretching. PPI and SPPH had a peak at 1392 and 1397 cm$^{-1}$, respectively [24]. A shift towards a higher wavenumber, as reported here, may represent a less coupled protein network with higher freedom-degree of its components, which has been attributed to a reduction of β-sheet interactions and an increase in disordered structures after the hydrolysis [21]. Generally, it is shown in Figure 3 that the overall structure of the protein fractions has not been modified by the treatment and that the intensity of SPPH is lower compared to PPI due to a lower concentration of SPPH sample [21].

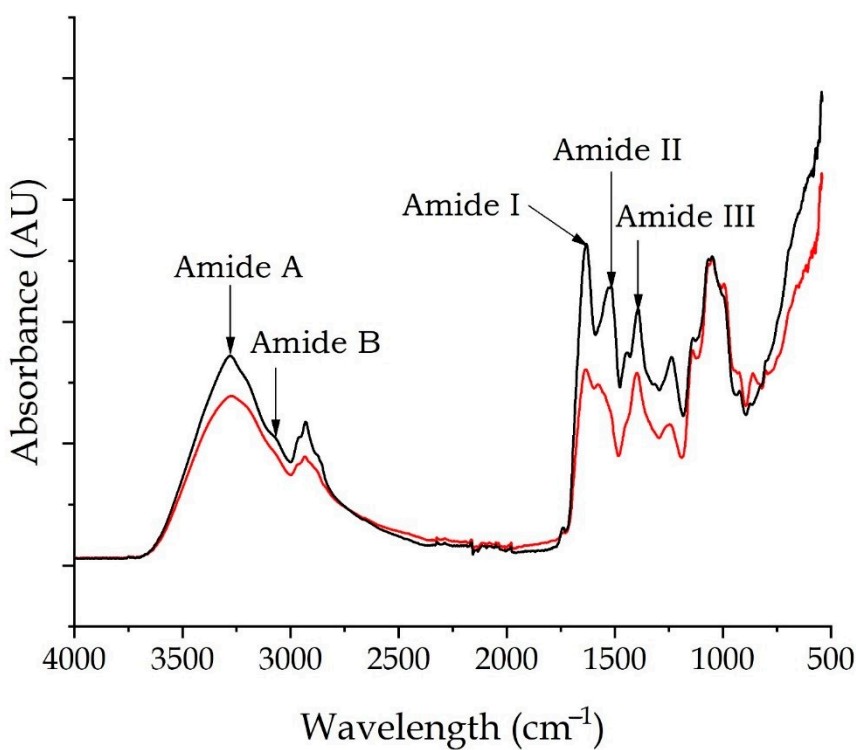

**Figure 3.** FTIR spectra of PPI (black line) and SPPH (red line), expressed as absorbance (AU) over wavelength (cm$^{-1}$).

### 3.3. Differential Scanning Calorimetry (DSC)

Differential scanning calorimetry thermogram provides information about phase transition, such as melting and glass transition, as well as protein denaturation. Melting occurs during the absorption of latent heat during the isothermal change solid–liquid, while glass transition is defined as the variations in the molecular mobility and relaxation time in amorphous solids [24]. Glass transition ($T_g$), protein denaturation ($T_d$), and melting point ($T_m$) of PPI and SPPH are shown in Figure 4. $T_g$ was not observed for PPI as previously reported by Vall-llosera et al. [25]. $T_d$ and $T_m$ of PPI were 155.3 $\pm$ 1.5 and 194.3 $\pm$ 1.2 °C, respectively. Similar $T_d$ was found by Sochava and Smirnova [26], who studied the denaturation of purified legumin. A different $T_m$ (180 °C) was reported by Jia et al. [27] which could be attributed to the different water content of the samples [26]. $T_g$ of SPPH was observed at 47.6 $\pm$ 2.8 °C. A similar value (41 °C) was reported by Pelgrom et al. [28] for pea protein isolate with dry matter up to 90%. $T_d$ and $T_m$ of SPPH were 162.4 $\pm$ 4.3 and 200.9 $\pm$ 5.7 °C, respectively. The high values obtained, both for PPI and SPPH, generally represent good thermal stability that was not reduced by the treatment.

Proof of a conformational change after the hydrolysis is given by the enthalpy difference ($\Delta H$). This value indicates the degree of order in a protein structure [29] and it is associated with the change of molecular structure during the unfolding [30]. $\Delta H$ is the combined effect of endothermic reactions, such as disruption of hydrogen bonds and exothermic reactions, such as the formation of hydrophobic bonds and/or protein aggregation [30]. Figure 4 showed that the melting peak for both PPI and SPPH was an endothermic peak, indicating that the disruption of the hydrogen bond was the main contributor. In this study, the enthalpy of melting ($\Delta H_m$) was found to be significantly lower for SPPH (49.5 $\pm$ 5.8 J/g) compared to PPI (81.2 $\pm$ 1.9 J/g) showing a modified tertiary structure of the proteins after the treatment. The same tendency was reported for hydrolyzed corn glutelin [31] and hydrolyzed wheat gluten [29]. Moreover, since a more flexible structure require less energy for melting [29], it can be suggested that enzymatic hydrolysis altered the ordered structure of the proteins into a more flexible one [31], where intermolecular distances increased due to higher repulsive forces.

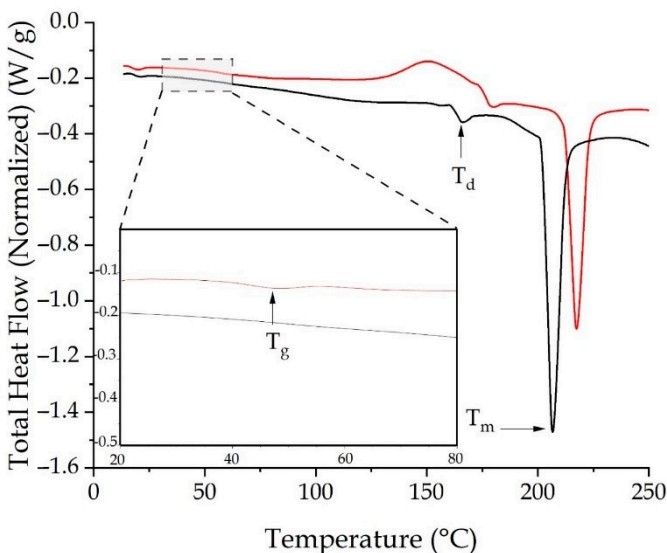

**Figure 4.** DSC chromatogram of PPI (black line) and SPPH (red line).

### 3.4. Hydrodynamic Diameter

Hydrodynamic diameter (*Dh*) of SPPH particles in solutions with different ionic strengths is presented in Figure 5. Before NaCl addition, SPPH exhibited a *Dh* of ~177 nm, while lower NaCl concentrations (0.1 and 0.2 M) significantly decreased the size to ~148 nm. The decrease might be associated with the salting-in effect, which can increase the solubility of the protein and thus inhibit molecular aggregation [13]. Salting-in effect of globular proteins (chymosin, β-lactoglobulin and pumpkin seed globulin) at 0.3 M NaCl was also reported by Maurer et al. [32]. On the contrary, at higher NaCl concentration (0.4 M), *Dh* increased up to ~190 nm. Similar behavior was observed by Tan et al. [13] for walnut protein, where *Dh* initially decreased until 0.15 M NaCl and then continuously increased until 0.5 M. According to the authors, these results can be attributed to the salting-out effect, which is responsible for molecular aggregation.

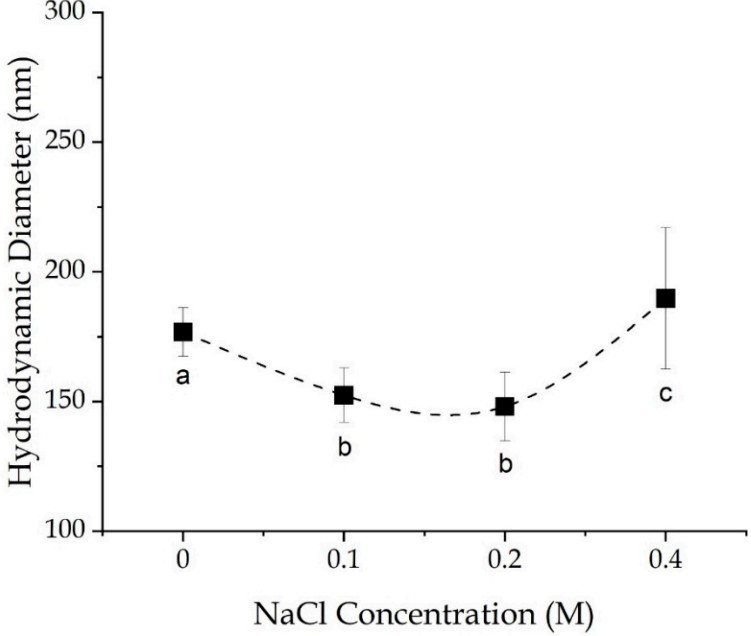

**Figure 5.** Hydrodynamic diameter (nm) of SPPH at different NaCl concentrations. Different letters in the line show significant difference ($p < 0.05$). Dash line is a guide to the eye.

### 3.5. Turbidity

Turbidity can be used as an additional qualitative method to evaluate the development and the size of aggregates [33]. Turbidity of SPPH solutions in different ionic strengths is presented in Figure 6 and it is expressed as absorbance at 400 nm. No remarkable difference was observed for samples between 0 and 0.2 M NaCl. These findings are in agreement with those reported by Munialo et al. [34] for pea protein solutions (150 mg/mL) and 0.3 M NaCl. Nevertheless, the turbidity increased with the addition of 0.4 M NaCl, which is probably attributed to the higher hydrodynamic diameter of the aggregates as it was measured by DLS (Figure 5).

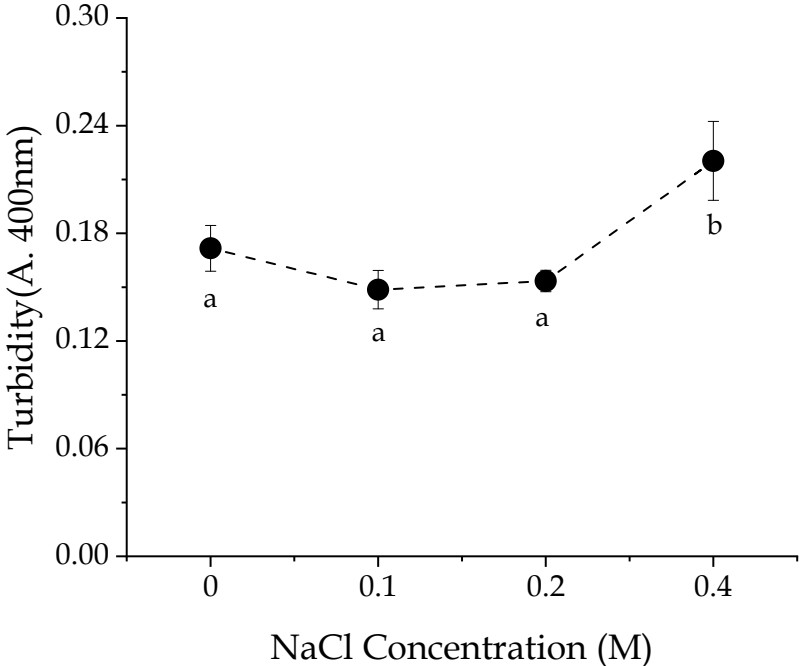

**Figure 6.** Turbidity of SPPH in different NaCl concentrations. Different letter indicates significant difference ($p < 0.05$). Dash line is guide to the eye.

### 3.6. Interfacial Pressure and Adsorption Kinetics

The effect of NaCl in adsorption of SPPH in the oil/water interface was monitored for 1800 s and the results are presented in Figure 7, where the interfacial pressure ($\pi$) was plotted over time. Adsorption kinetics parameters such as diffusion rate ($k_{diff}$), penetration rate ($k_p$) and rearrangement rate ($k_r$), were determined by the plot of Equation (4) and are represented in Table 2.

$\pi$ values close to zero were not detected, indicating a good surface activity of SPPH. The addition of NaCl improved $\pi$ value in all concentrations and the highest $\pi_{1800}$ value (14.28 mN/m) was measured at 0.4 M NaCl. Similar findings were described by Tan et al. [13] for the walnut protein-xanthan gum system at various NaCl concentrations (0.05–0.5 M). The increase in interfacial pressure was ascribed to the counter-ion screening effect, which provoked a decrease in surface charge and consequently an increase in protein hydrophobicity [2,13].

The estimation of $k_{diff}$ during the initial step was limited by the experimental method used in this report, however, the initial jump in $\pi$ (Figure 7) is a qualitative measure of the diffusion of proteins [35]. It could be noticed that at 0.4 M NaCl the initial jump was higher compared to lower NaCl concentrations. In accordance with these results, Ruíz-Henestrosa et al. [36] found that the increase in ionic strength from 0.05 M to 0.5 M increased $k_{diff}$ of soy vicilin at pH 7 by 4-fold.

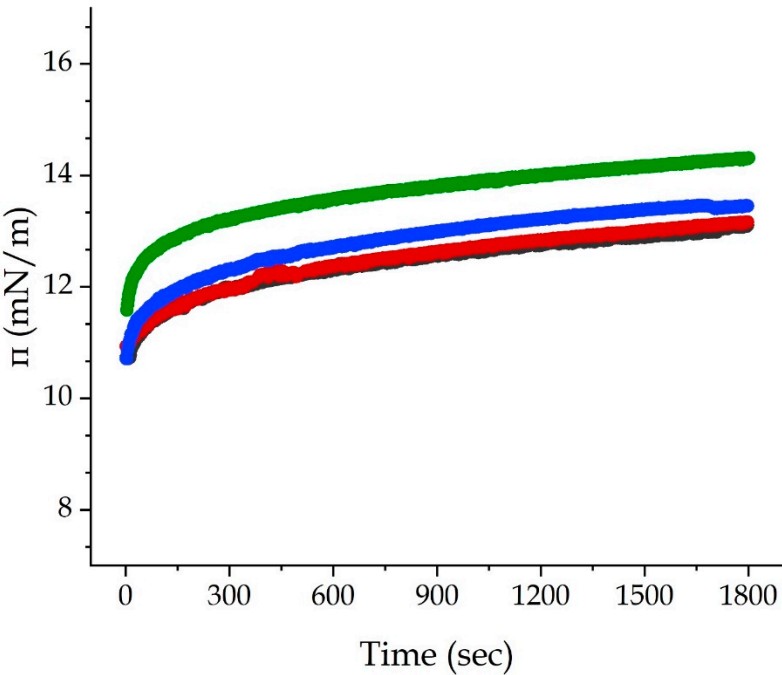

**Figure 7.** Interfacial pressure of SPPH at the water/oil interface at 20 °C, pH 7.2, and at 0 (grey line), 0.1 (red line), 0.2 (blue line) and 0.4 M (green line) NaCl over 1800 s.

**Table 2.** Characteristic parameters for the adsorption of SPPH (0.5%) at the water/oil interface at 20 °C, pH 7.2, and varying salt concentration. [a] LR = linear regression coefficient.

| NaCl (M) | $\pi_{1800}$ (mN/m) | $k_p \times 10^4$ (s$^{-1}$) (LR) [a] | $k_r \times 10^4$ (s$^{-1}$) (LR) [a] |
|---|---|---|---|
| 0 | 13.03 ± 0.06 [a] | 13.9 (0.995) | 29.2 (0.920) |
| 0.1 | 13.11 ± 0.03 [b] | 15.2 (0.989) | 52.9 (0.958) |
| 0.2 | 13.43 ± 0.14 [c] | 15.6 (0.993) | 30.7 (0.974) |
| 0.4 | 14.28 ± 0.03 [d] | 14.4 (0.995) | 31.1 (0.970) |

Significance difference among the values is marked with different letters and was obtained by one-way ANOVA and Tukey's test at $p < 0.05$.

$k_p$ and $k_r$ were fitted well to the first-order equation (LR > 0.958) for all the samples with salt. The addition of small amounts of NaCl (0.1–0.2 M) improved both penetration and rearrangement of SPPH. Maximum value of $k_p$ was $15.6 \times 10^4$ s$^{-1}$ for 0.2 M NaCl and of $k_r$ was $52.9 \times 10^4$ s$^{-1}$ for 0.1 M NaCl. At a neutral pH and low salt concentration (0.1 M), electrostatic repulsions between protein molecules could cause exposure of hydrophobic groups and enhance the penetration at the interface [37]. At 0.4 M NaCl, both $k_p$ and $k_r$ decreased compared to 0.1 M NaCl. At higher salt concentrations (0.4 M NaCl), the reduction of electrostatic repulsions promotes a denser packing of protein molecules [38].

Considering the adsorption kinetics of SPPH in an oil/water interface, it can be concluded that diffusion is the main mechanism that controls protein adsorption [36]. Penetration is correlated with NaCl concentration (Table 2); however, no correlation was found between NaCl concentration and the rearrangement rate, suggesting that post-adsorption changes are not influenced by ionic strength [38].

### 3.7. Dilatational Interfacial Properties

Dilatational storage (E′) and loss (E″) moduli were used to describe the viscoelastic properties of the interfacial film in different frequencies (0.1, 0.2, 0.4 and 1 Hz) and are presented in Figure 8. E′ is an indication of the strength of the interfacial film, formed by protein interactions owing to conformational rearrangement and of the film rigidity, responsible for deformation resistance at the interface [7]. On the contrary, E″ represents the

energy loss through the relaxation process [7]. All samples displayed dominant E′ for all the examined frequencies (Figure 8), implying the formation of a viscoelastic interfacial film. Moreover, loss factor tan $\delta$ (E″/E′) and its evolution with frequency and salt concentration was obtained (Supplementary File Figure S1). Values below or equal to 0.6 were obtained for all the frequencies considered, indicating the prevalent elasticity of the interfacial membrane [13].

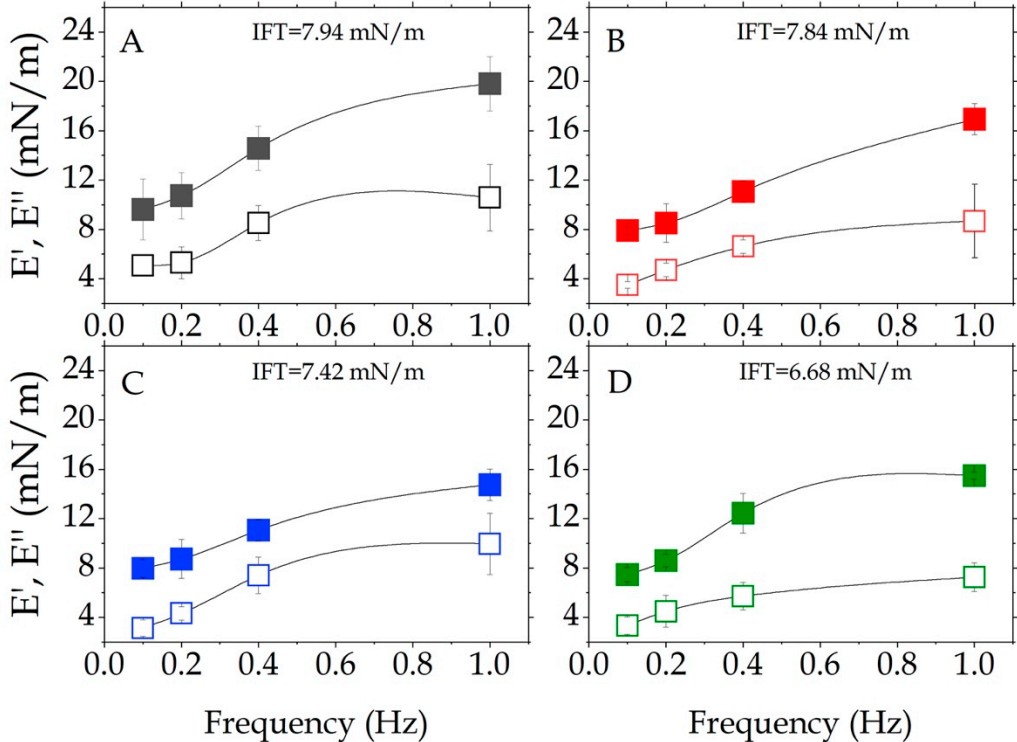

**Figure 8.** Interfacial viscoelastic properties of SPPH at room temperature (20 ± 0.01 °C) with 0 M (**A**), 0.1 M (**B**), 0.2 M (**C**) and 0.4 M (**D**) NaCl. Storage modulus (E′) represented with closed symbols and loss modulus (E″) with open symbols. Final Interfacial Tension Values (IFT) are also presented for each sample. Dash line is guide to the eye.

For all samples, no significant frequency dependency was observed at lower frequencies (0.1 and 0.2 Hz) for both E′ and E″, indicating a very slight relaxation process [7]. A similar trend was reported by Won et al. [39] for β-lactoglobulin at water/tetradecane interface, at frequencies from 0.001 to 1 Hz and low ionic strength (1–100 mM). At 0.4 Hz, the sample without NaCl exhibited the highest E′, while the addition of 0.1 and 0.2 M NaCl provoked a decline from 14.56 to 11.06 mN/m and an increase from 0.58 to 0.6 and 0.67 of tan $\delta$ values. However, E′ increased with further addition of NaCl up to 0.4 M, from 11.06 to 12.45 mN/m, while at the same time tan $\delta$ decreased from 0.67 to 0.46, reaching an even lower value than the sample without salt. Similar behavior was also observed at 1 Hz, where E′ initially decreased from 19.80 to 14.74 mN/m and tan $\delta$ rised from 0.53 to 0.68 (0–0.2 M NaCl), while subsequently E′ increased from 14.74 to 15.47 mN/m and tan $\delta$ diminished to 0.47 (0.4 M NaCl). No significant difference was observed for E″ at 0.4 and 1 Hz for all the samples. This particular behavior is in agreement with the results obtained by DLS where a small addition of salt manifested the salting-in effect, which reduced the interactions between the molecules, forming smaller particle aggregates and thus inhibiting the formation of a strong viscoelastic layer at the interface. On the contrary, an increase in salt strength (0.4 M NaCl) caused the salting-out effect, which improved peptide interactions, formed aggregates with larger hydrodynamic diameter and thus a stronger adsorbed membrane. Moreover, the evolution of E′ along with increasing NaCl concentration is in perfect accordance with $k_p$ parameter. Indeed, high penetration at low

salt concentration is an indicator of a weak interfacial viscoelastic layer, while a lower penetration at a high salt concentration indicated a strong interfacial viscoelastic layer. Therefore, NaCl influences the peptides unfolding and film forming ability at the interface.

However, a lower dilatational storage value as well as a higher $k_p$ value have been obtained at 0.4 M when compared to the same values of the control sample without any salt addition. Therefore, the addition of ionic strength had a negative effect on the viscoelastic properties of the interfacial film formed by SPPH. These results are in general accordance with the study by Tan et al. [13]. The authors found that, the complex of walnut proteins and xantham gum considered, presented a decline in E′ from 48.47 to 34.19 mN/m and a rise in tan $\delta$ from 0.11 to 0.16 when NaCl concentration was increased from 0.15 to 0.35 M, while with higher concentration of NaCl up to 0.5 M, E′ raised from 34.19 to 38.01 mM/m and tan $\delta$ declined from 0.16 to 0.12, manifesting also in this case a lower dilatational storage value, compared to the control without any salt addition and thus a negative effect on the viscoelastic properties at the interface.

*3.8. Shear Interfacial Rheology*

An interfacial time sweep was conducted for 25 h at a frequency of 0.2 Hz and a 0.1% strain amplitude to monitor the adsorption of hydrolysed peptides at the oil/water interface (Figure 9). The storage modulus (G′), indicating elastic strength, the loss modulus (G″), indicating viscous behavior, and the loss tangent tan $\delta$ (G″/G′) values were determinated.

Three distinct phases of peptides absorption can be found by a time sweep measurement. The first phase is diffusion, which is governed by the bulk temperature, the particle size and the net charge. The duration of this phase can be established as the time in which the rheometer can detect a reliable elastic modulus (G′> $10^{-4}$ N/m) [38]. To overcome this induction period, high peptides concentration was used. The second phase includes peptide rearrangement at the interface and the formation of a monolayer. It can be identified as a steep increase of G′ and a corresponding decrease of tan $\delta$ [38]. The third phase is referred to as the steady state, where G′ and tan $\delta$ reach a plateau [15].

All the samples demonstrated predominant elastic behavior (G′ > G″), however different crossover points were observed. Particularly, the sample without salt displayed a crossover point at 5.4 h, while the addition of 0.1 and 0.2 M NaCl resulted in a crossover point at 6.5 and 7.2 h, respectively. At 0.4 M NaCl, the time required for G′ > G″ reduced at 6.3 h. After that point, a rapid increase of G′and a decrease of tan $\delta$ were found for samples with 0 and 0.4 M NaCl, on the contrary, the rise and the fall of G′ and tan $\delta$, respectively, were slower for samples with 0.1 and 0.2 M NaCl. Moreover, samples with 0.1 and 0.2 M NaCl reached steadier G′ values after 18 and 21 h, respectively, while no steady values were reached for 0 and 0.4 M NaCl within the monitoring time but kept on rising. However, when looking at the tan $\delta$ values evolution with time, the situation is the opposite: 0 and 0.4 M samples presented constant values after 10 h, while 0.1 and 0.2 M samples kept on oscillating in tan $\delta$ values within the overall period of time considered.

The different times of the moduli crossover and the evolution of G′ and tan $\delta$ values over time reflect precisely the effect of the salt on the folding and rearregments of the peptides mixture and thus on their intermolecular interactions, which are the main responsible for the formation of the interfacial layer membrane. Once again in fact, a small salt addition (0.1–0.2 M) increased the elestrostatic repulsions of the soluble particles, while the higher concentration level of 0.4 M promoted their agglomerization at the interface.

After 25 h, G′ values of 84, 5, 6 and 25 mN/m and tan $\delta$ of 0.11, 0.25, 0.17 and 0.12 were obtained for samples with 0, 0.1, 0.2 and 0.4 M NaCl, respectively. It can be seen that, in accordance with the dilatational properties results (Figure 8), the presence of NaCl had in general a negative effect on the strength of the interfacial layer. Nevertheless, among the samples with NaCl, an increase in salt concentration, and thus ionic strength, provoked an improvement of G′ values as well as a reduction of the tan $\delta$ values in function of time, indicating a prevailing elastic behavior, an improved intermolecular interaction and a more compact interfacial membrane. This may be ascribed to the screening effect, which promotes

hydrophobic interactions between peptides and the formation of aggregates [13,40]. This hypothesis is also supported by the hydrodynamic diameter of the particles as it was measured by DLS (Figure 5).

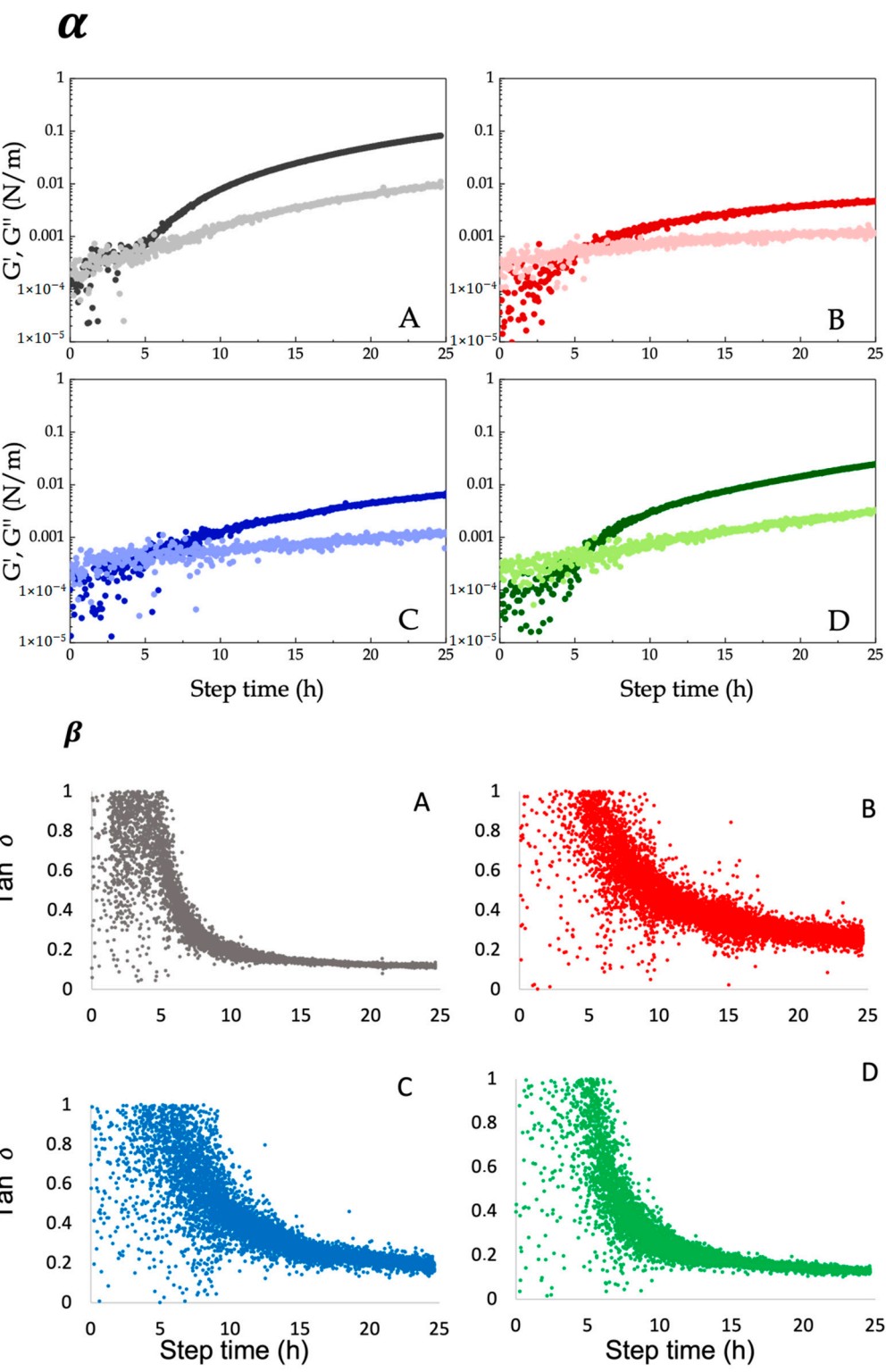

**Figure 9.** Time sweep for 25 h at 0.2 Hz, 0.1% strain and 20 °C. ($\alpha$) Storage G$'$ (dark colored) and loss G$''$ (light colored) moduli; ($\beta$) tangent loss tan $\delta$ of SPPH samples with 0 (**A**), 0.1 (**B**), 0.2 (**C**) and 0.4 (**D**) M NaCl are here presented.

## 4. Conclusions

Partial tryptic hydrolysis of pea protein significantly reduced the size of polypeptide chains and also affected their tertiary structures due to the disruption of hydrogen bonds. In a solution system, a salting-in effect was observed with the addition of 0.1 and 0.2 M NaCl, as the hydrodynamic diameter decreased, however, turbidity was not affected. On the contrary, salting-out effect occurred at 0.4 M of NaCl, inducing the formation of larger agglomerates with a higher hydrodynamic diameter that increased the turbidity of the medium. In an oil/water interfacial environment, the addition of NaCl improved the surface activity of SPPH, due to the counter-ion screening effect. Nonetheless, dilatational and interfacial shear properties were negatively affected by the addition of NaCl, which induced changes in the intermolecular interactions between the adsorbed peptides. However, at high salt concentrations, the elasticity and the strength of the interfacial layer improved when compared to lower salt concentrations. This has been attributed to the formation of better packed aggregates at the interface. It can be concluded that changes in ionic strength significantly affect the bulk and the interfacial properties of SPPH. This study monitored the effect of SPPH as a potential techno-functional ingredient, which could be employed in the food industry for the design and development of different products, such as beverages, emulsions and savory sauces, for example. For instance, the stability and characteristics of such products are highly dependent on their bulk and interfacial properties. Properties that, in this study, for SPPH were further evaluated in the presence of different salt strengths, highlighting that it could potentially be employed as surface-active ingredient in plant-based food products with the ability to modify interfacial functionalities at the desired characteristics.

**Supplementary Materials:** The following supporting information can be downloaded at: https://www.mdpi.com/article/10.3390/colloids6040076/s1, Table S1: Z-potential and Z-size values for all the samples with statistical interpretation, Table S2: DLS analysis and test parameters and conditions, Table S3: Frequency dependence of dilatational viscoelastic properties of the samples, Table S4: tangent loss factor values for dilatational viscoelastic properties, Figure S1: Tangent loss factor evolution in function of salt concentration for all the considered frequencies; Figure S2: Particle size distribution for SPPH sample with 0 M NaCl concentration; Figure S3: Interfacial Tension Values over time of all of the samples.

**Author Contributions:** Conceptualization, K.S. and D.O.; methodology, K.S.; validation, F.C. and F.J.; formal analysis, A.F.d.C., M.V.-l. and F.A.; investigation, K.S.; data curation, K.S. and D.O.; writing—original draft preparation, K.S. and D.O.; writing—review and editing, D.O. and F.C.; project administration, A.F.d.C.; Mohammadifar M.A.M.; funding acquisition, F.C. All authors have read and agreed to the published version of the manuscript.

**Funding:** The authors acknowledge the support provided by the Green Development and Demonstration Program (GUDP) and Ministry of Environment and Food of Denmark (J. Nr. 34009-17-1299).

**Data Availability Statement:** Data is contained within the article or the supplementary material.

**Acknowledgments:** The authors acknowledge Roquette® for kindly providing the pea protein sample. Data was generated through accessing research infrastructure at DTU National Food Institute, including FOODHAY (Food and Health Open Innovation Laboratory, Danish Roadmap for Research Infrastructure).

**Conflicts of Interest:** The authors declare that there are no conflict of interest.

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
