# Peer review of "Interfacial Properties of Pea Protein Hydrolysate: The Effect of Ionic Strength"

_colloids, doi:10.3390/colloids6040076_

Round 1
Reviewer 1 Report
Article Ref. No: Colloids 1959715
General comments:
The manuscript studies the structural changes of pea protein after hydrolysis and the ionic strength on oil/water interfacial properties of the hydrolysates to develop of food colloids with diverse functionality based on vegetal ingredients. The manuscript contains interesting data, however FTIR and viscoelastic analysis should be changed and improved.
Specific comments
1) Line 17: indicate the specific temperature. Line 86: are the equivalents?
2) Lines 240-241: “maximum absorbance at 1632 cm-1 confirming that the secondary structure of the proteins involved wasn’t modified by the hydrolysis treatment” this statement cannot be deduced from FTIR spectra of Figure 3. It is needed to obtain the deconvoluted amide I band between 1600 and 1700 cm-1 to analyse the possible formation of non-native ordered structure such as the hydrogen-bonded beta sheets around 1620 cm-1 which may be correlated with protein aggregation.
3) Lines 246-247: “A shift towards higher wave-number, is an indication of a weaker protein network” this statement is confuse since a higher wavenumber is equivalent to higher frequency, thus a greater energy content of vibration bond. Therefore, if there is a shift to higher frequencies, bonds are stronger (greater energy content). It is possible that higher wavenumber indicates a less coupled network with higher number of vibration freedom-degrees. A less coupled structure is different than a weaker network. For that reason it is important to obtain the deconvoluted amide I band.
4) Line 250: “The peak at 1234 cm-1 is linked with C-H bond of protein” revise! the stretching band of C-H is around 2900 cm-1.
5) Line 284-285: “enzymatic hydrolysis altered the ordered structure of the proteins into a more flexible structure”. Authors should distinguish between thermal and mechanical effects. So, flexibility is a mechanical effect based on the strand curvature in network. From DSC data Authors should reason in terms of energy implications associated to the hydrolysis products which increase the number of charged groups (NH4+ and COO- ) and the subsequent increase in the repulsive forces (higher molecular distances) compatible with the lower ΔHm for SPPH. Authors speculate about flexibility without analyse the curvature of strands.
6) Line 357: the experimental conditions of Figure 8 should be included in material and methods section. Frequency sweeps? Time sweeps? In Figure caption (Fig. 8) should be included the temperature.
7) Lines 367-372: introduce the experimental errors of different amounts to improve the explanation.
8) Lines 372-377: Authors should introduce the loss factor values (tand=E’’/E’) to compare the effect of NaCl concentration with frequency on the net viscoelasticity. Idem for lines 384-386, Authors should reason in terms of the tand as a function of salt content and frequency. So, the viscoelastic properties at the interface should also be explained considering the tand values, because they provide the degree of viscoelaticity.
9) Lines 401-402: These lines contain similar information with previous paragraphs (358-360). Remove!
10) Line 406: The units of G’ and G’’ in the IS are Pa (N/m2). Revise the units of magnitudes (y-axis) for Figures 8 and 9.
11) Lines 416-417: “samples with 0.1 and 0.2 M NaCl reached a plateau after 18 and 21 h” I think that there is not a true plateau, since it may be observed a slight increase of G’. Authors should obtain the time dependence of tand for each sample between 15 and 25 h, to compare the increase in the viscoelastic degree among samples.
12) Lines 425-428: Authors should compare tand values with increasing time, to give a more complete view of the viscoelastic response for all samples. They analyse the strength in terms of G’, this is a partial view of the viscoelastic analysis. So, the more relevant is the relation between both G’’ and G’ with increasing time. The same suggestion for lines 427- 432.
13) Lines 447-448: “the elasticity and the strength of the interfacial layer improved when compared to lower salt concentrations” this sentence should be assessed considering tand values.
14) Table 2: Write at bottom place methodology for obtaining letters of significance for P values.
Author Response
Dear Editor,
Please find below the answers to the reviewers’ comments. When changes have been made in the article, they have been precisely reported to the reviewers. To facilitate the revision, all the changes in the text are in red.
Best regards,
Federico Casanova and Collaborators.
Reviewer 1
General comments:
The manuscript studies the structural changes of pea protein after hydrolysis and the ionic strength on oil/water interfacial properties of the hydrolysates to develop of food colloids with diverse functionality based on vegetal ingredients. The manuscript contains interesting data, however FTIR and viscoelastic analysis should be changed and improved.
Specific comments
1) Line 17: indicate the specific temperature. Line 86: are the equivalents?
The temperature at which the analysis was done was indeed missing; temperature value has thus been specified in the abstract and the correction from is to are, line 86, has been applied.
2) Lines 240-241: “maximum absorbance at 1632 cm-1 confirming that the secondary structure of the proteins involved wasn’t modified by the hydrolysis treatment” this statement cannot be deduced from FTIR spectra of Figure 3. It is needed to obtain the deconvoluted amide I band between 1600 and 1700 cm-1 to analyse the possible formation of non-native ordered structure such as the hydrogen-bonded beta sheets around 1620 cm-1 which may be correlated with protein aggregation.
We appreciate the valuable comment of the reviewer and for this reason the statement has been reconsidered into an hypothesis of the unchanged secondary structure of the peptides after the applied treatment. In this way it will be clearer for the reader and will not be confused.
3) Lines 246-247: “A shift towards higher wave-number, is an indication of a weaker protein network” this statement is confuse since a higher wavenumber is equivalent to higher frequency, thus a greater energy content of vibration bond. Therefore, if there is a shift to higher frequencies, bonds are stronger (greater energy content). It is possible that higher wavenumber indicates a less coupled network with higher number of vibration freedom-degrees. A less coupled structure is different than a weaker network. For that reason it is important to obtain the deconvoluted amide I band.
We appreciate the valuable comment of the reviewer regarding this particular sentence. The indicated statement has therefore been revised and modified in order to appear clearer without space for any misinterpretation of it. The difference between weaker and less coupled network has been understood and implemented into the manuscript.
4) Line 250: “The peak at 1234 cm-1 is linked with C-H bond of protein” revise! the stretching band of C-H is around 2900 cm-1.
We understand the reviewer however, we decided to delete the sentence.
5) Line 284-285: “enzymatic hydrolysis altered the ordered structure of the proteins into a more flexible structure”. Authors should distinguish between thermal and mechanical effects. So, flexibility is a mechanical effect based on the strand curvature in network. From DSC data Authors should reason in terms of energy implications associated to the hydrolysis products which increase the number of charged groups (NH4+ and COO- ) and the subsequent increase in the repulsive forces (higher molecular distances) compatible with the lower ΔHm for SPPH. Authors speculate about flexibility without analyse the curvature of strands.
Indeed, we have not considered the curvature of the strands, basing our discussion only on the thermal effects of the applied analysis. The implied sentence was only based on the discussion of the results of the considered references [29, 31]. In here in fact, the same tendency was described by the authors for different hydrolyzed proteins, such as corn glutelin and wheat gluten, which used the concept of flexibility to explain their results. We understand and agree that this “flexibility” in reality represents the increase in repulsive forces, responsible for higher intermolecular distances which has also been clarified into the manuscript.
6) Line 357: the experimental conditions of Figure 8 should be included in material and methods section. Frequency sweeps? Time sweeps? In Figure caption (Fig. 8) should be included the temperature.
The experimental conditions of figure 8 were already described into the material and methods, under the section 2.9 “interfacial pressure and interfacial dilatational properties”. However it has been now clarified that the tests applied were frequency sweep tests with fixed amplitude as indicated by the reviewer. Temperature of the test has been also added into the caption of figure 8.
7) Lines 367-372: introduce the experimental errors of different amounts to improve the explanation.
We agree with the reviewer and we include the std.
8) Lines 372-377: Authors should introduce the loss factor values (tand=E’’/E’) to compare the effect of NaCl concentration with frequency on the net viscoelasticity. Idem for lines 384-386, Authors should reason in terms of the tand as a function of salt content and frequency. So, the viscoelastic properties at the interface should also be explained considering the tand values, because they provide the degree of viscoelaticity.
Please, see below the tand calculation based on the average E’ and E’’.
|
Salt Concentration |
Frequency |
E' (average) |
std |
E'' (average) |
std |
tand |
|
M |
Hz |
mN/m |
mN/m |
|||
|
0 |
0.1 |
9.61975 |
2.46036 |
5.08263 |
0.88275 |
0.53 |
|
0 |
0.2 |
10.72588 |
1.88082 |
5.29875 |
1.294 |
0.49 |
|
0 |
0.4 |
14.5634 |
1.79493 |
8.514 |
1.42339 |
0.58 |
|
0 |
1 |
19.801 |
2.198 |
10.573 |
2.68971 |
0.53 |
|
0.1 |
0.1 |
7.9068 |
0.28535 |
3.5076 |
0.27656 |
0.44 |
|
0.1 |
0.2 |
8.5116 |
1.57073 |
4.7065 |
0.54741 |
0.55 |
|
0.1 |
0.4 |
11.0582 |
0.85369 |
6.613 |
0.53446 |
0.60 |
|
0.1 |
1 |
16.92233 |
1.26573 |
8.67025 |
2.97983 |
0.51 |
|
0.2 |
0.1 |
7.9634 |
0.71505 |
3.126 |
0.66574 |
0.39 |
|
0.2 |
0.2 |
8.7298 |
1.57165 |
4.321 |
0.54683 |
0.49 |
|
0.2 |
0.4 |
11.0582 |
0.85369 |
7.4056 |
1.47573 |
0.67 |
|
0.2 |
1 |
14.739 |
1.28494 |
9.95033 |
2.47262 |
0.68 |
|
0.4 |
0.1 |
7.46383 |
0.5794 |
3.32683 |
0.69668 |
0.45 |
|
0.4 |
0.2 |
8.598 |
0.49679 |
4.482 |
1.2911 |
0.52 |
|
0.4 |
0.4 |
12.4348 |
1.59882 |
5.7188 |
1.11906 |
0.46 |
|
0.4 |
1 |
15.474 |
0.28526 |
7.2476 |
1.16041 |
0.47 |
9) Lines 401-402: These lines contain similar information with previous paragraphs (358-360). Remove!
We have not notice that repetition, we just wanted to make clear for the readers the meaning of the considered and studied variables. Nonetheless, the repetition has been detected and consequently removed as suggested by the reviewer.
10) Line 406: The units of G’ and G’’ in the IS are Pa (N/m2). Revise the units of magnitudes (y-axis) for Figures 8 and 9.
We decided to collect and maintain values in mM/m and N/m for E and G moduli respectively, in order to simplify the reader comprehension of their trends with different salt concentrations. The values were in fact compared in the discussion to other references [13,38] who collected and presented the results in the same units. In this way, the reader could easily assess the similar trend and compare the values of the moduli for different hydrolysed protein.
11) Lines 416-417: “samples with 0.1 and 0.2 M NaCl reached a plateau after 18 and 21 h” I think that there is not a true plateau, since it may be observed a slight increase of G’. Authors should obtain the time dependence of tand for each sample between 15 and 25 h, to compare the increase in the viscoelastic degree among samples.
Indeed, we agree with the reviewer that a slight increase in G’ values is still visible. However, compared to the increase shown before 18 and 21h for the sample with 0.1 and 0.2 M NaCl it can be described as a steadier state. The slight differences in fact may have also been induced by the unavoidable disturbances on the instrument during the long period of time considered. It is indeed impossible to maintain a steady environment around a very sensible instrument, as the one employed, for these analysis for a long period of time. Nonetheless, the affirmation in the manuscript has been corrected to highlight for the reader that steadier values have been detected after those time points for the considered samples, while the same could not be said for the 0 and 0.4 M NaCl samples.
12) Lines 425-428: Authors should compare tand values with increasing time, to give a more complete view of the viscoelastic response for all samples. They analyse the strength in terms of G’, this is a partial view of the viscoelastic analysis. So, the more relevant is the relation between both G’’ and G’ with increasing time. The same suggestion for lines 427- 432.
13) Lines 447-448: “the elasticity and the strength of the interfacial layer improved when compared to lower salt concentrations” this sentence should be assessed considering tand values.
14) Table 2: Write at bottom place methodology for obtaining letters of significance for P values.
We agree that statistical information was missing in table 2 caption and we have proceeded in indicating the methodology used to obtain it.

Reviewer 2 Report
The article “Interfacial properties of pea protein hydrolysate: The effect of ionic strength” was submitted to Colloids and Interfaces for the special issue “Food Colloids II”.
General observations:
In this manuscript, the authors investigated the changes in the structure and size of pea protein after tryptic hydrolysis and the effect of ionic strength on oil/water interfacial properties of the hydrolysates. After a brief introduction concerning the topic and the related problematic, the body of the article is divided into 8 parts, each corresponding to a method of characterization in the following order : molecular weight distribution (SEC-MALS & SDS-PAGE), FTIR, DSC, DLS, turbidity, interfacial pressure, interfacial dilatational properties, and shear interfacial rheology.
The English language and style are correct, only minor spelling are required.
The article is well documented, there are many references and the ‘Materials and Methods’ section is well detailed. However, some parts of the ‘Results and discussion’ need more detail and elaboration. Although the authors make comparisons in the results section with other results obtained from literature, a section is missing before the conclusion, gathering the different properties and justifying that SPPH is a good candidate for food applications. Therefore, for all of the above reasons and my many comments that follow, I cannot recommend publication of this work in Colloids and Interfaces before a major revision.
Observations and comments:
2. Materials and Methods:
- 2.7 Hydrodynamic diameter :
Page 3 line 141-142 : The authors should clarify what they mean by "hydrodynamic diameter". Is it the intensity-weighted mean hydrodynamic size? What about the polydispersity index (PdI)?
Page 4 line 146 to 151 : The authors did not specify the viscosity value and the measurement temperature used to calculate the hydrodynamic diameter.
- 2.8 Turbidity :
Turbidity measurements were made with a UV-vis spectrophotometer, so it would be more appropriate to speak of absorbance to avoid confusion. It would also be better for the understanding to add the full absorbance spectrum to verify that the chosen wavelength (400 nm) was correct. This curve could be added as supporting information.
- 2.9 Interfacial pressure and interfacial dilatational properties :
Please add the formula of interfacial pressure.
Pendant drop tensiometry requires to know the density of each phase studied, however this information does not appear here.
The oscillatory tests were performed in 4 steps with 10 s waiting between each cycle. Could the authors justify why they chose this value of 10 s between each cycle of oscillations? Is this time sufficient for the interface to relax?
3. Results and discussion
- 3.4 Hydrodynamic diameter :
The authors directly reported Dh (Hydrodynamic diameter) as a function of salt concentration. However, when characterizing the size of a dispersion one should always plot the size distribution of at least one of the systems to get a better idea of the distribution. This can be done here as supporting information.
Do the error bars in Fig.5 come from the standard deviation of the triplicate or from the polydispersity index? Could you comment about the error bar which is much higher for the 0.4 M NaCl concentration?
Have you considered that when the salinity of the water increases the viscosity of the continuous phase also increases? Admittedly, for a concentration of 0.4 M NaCl (23 g/L) the difference is small but the hydrodynamic diameter being directly inversely proportional to the viscosity, an increase of 10-20% of the viscosity will have a significant effect on the diameter.
The above points should be addressed to obtain a better interpretation of the size results.
- 3.5 Turbidity :
As a general rule, the smaller the particles, the more light with a shorter wavelength is scattered; and the larger the particles, the more light with a longer wavelength is scattered. The authors did not report the adsorption spectrum in the article, so it is difficult to confirm the interpretation made in this section. Is there a peak in absorbance?
- 3.6 Interfacial pressure and adsorption kinetics :
The concentration of the amphiphilic species should appear in this part.
The control (interfacial pressure without amphiphilic species) is missing in Fig.7 in order to estimate the effect of SPPH on the water/oil interfacial tension.
Can you justify the choice of 1,800 s to determine the interfacial pressure?
Line 338 p.11 : “Kp and kr were fitted well to the first order” : none of the fits appear in the article, could be added in supporting information.
- 3.7 Dilatational interfacial properties :
Page 12 lines 361-362 : add a reference.
E' and E" must always be compared to the value of the interfacial tension. Although all measured samples displayed dominating E' for all examined frequencies, if E' remains lower than the interfacial tension then the viscoelastic behavior will be weak. The value of the interfacial tension should be added for all systems in Fig.8.
Moreover, the authors did not add the curves of E' and E" as a function of time. Their temporal evolution is always a good indicator of the viscoelastic behavior. At least 1 curve should be added for each salt concentration at a given oscillation frequency, this can be done in supporting information.
Did the authors verify that there were no wrinkles on the surface of the drop during the oscillations?
Could you comment on the fact that the error bars reported at 1 Hz for all systems are much higher than for other frequencies?
- 3.7 Shear interfacial rheology :
It is not possible to differentiate G' from G" in Fig.9. Moreover, there is a lot of noise for all data reported for times less than 10 h. It is therefore impossible to identify for which times there is a crossing of the curves of G' and G". I recommend that the authors add the fits of each curve.
4. Conclusion:
Most of the conclusions drawn from the size, turbidity, and interfacial behavior study do not seem obvious and may need revision based on the suggested modifications.
The authors concluded “SPPH could be employed as clean label surfactants to improve foods turbidity, thickness and interfacial properties, allowing to design innovative products, such as beverages emulsions and savory sauces, with high plant-based protein content and desired technological characteristics” (lines 453-456). I don't see how the results justify that SPPH is a good candidate for the different applications mentioned, according to me it misses a comparative section in the article with another amphiphilic compound used in certain desired applications.
Author Response
Reviewer 2
General observations:
In this manuscript, the authors investigated the changes in the structure and size of pea protein after tryptic hydrolysis and the effect of ionic strength on oil/water interfacial properties of the hydrolysates. After a brief introduction concerning the topic and the related problematic, the body of the article is divided into 8 parts, each corresponding to a method of characterization in the following order : molecular weight distribution (SEC-MALS & SDS-PAGE), FTIR, DSC, DLS, turbidity, interfacial pressure, interfacial dilatational properties, and shear interfacial rheology.
The English language and style are correct, only minor spelling are required.
The article is well documented, there are many references and the ‘Materials and Methods’ section is well detailed. However, some parts of the ‘Results and discussion’ need more detail and elaboration. Although the authors make comparisons in the results section with other results obtained from literature, a section is missing before the conclusion, gathering the different properties and justifying that SPPH is a good candidate for food applications. Therefore, for all of the above reasons and my many comments that follow, I cannot recommend publication of this work in Colloids and Interfaces before a major revision.
Observations and comments:
- Materials and Methods:
- 2.7 Hydrodynamic diameter :
Page 3 line 141-142 : The authors should clarify what they mean by "hydrodynamic diameter". Is it the intensity-weighted mean hydrodynamic size? What about the polydispersity index (PdI)?
We agree that a clear definition of this variable was missing in the relative paragraph. For this reason, a definition of hydrodynamic diameter has been added at the beginning of the paragraph and the description of its equation has been improved to make the mean of this variable clear for the reader. Regarding the PdI, all the measurements here collected and reported have been considered significant and valid only if a PdI less than 0.5 was obtained by the instrument. Moreover, the instrument software was indicating the reliability of each result after the measurement. All this information has been added into the manuscript as suggested by the reviewer.
Page 4 line 146 to 151 : The authors did not specify the viscosity value and the measurement temperature used to calculate the hydrodynamic diameter.
We did not included information regarding values of temperature and viscosity because they were automatically assessed and took in consideration by the instrument and its calculating software to obtain the values using the equation 2, described into the relative paragraph.
- 2.8 Turbidity :
Turbidity measurements were made with a UV-vis spectrophotometer, so it would be more appropriate to speak of absorbance to avoid confusion. It would also be better for the understanding to add the full absorbance spectrum to verify that the chosen wavelength (400 nm) was correct. This curve could be added as supporting information.
We decided to present the results of turbidity not only considering the absorbance values obtained at 400 nm but further computing them into a simple and easy to understand graph over the concentration of NaCl for all the samples considered. In this way it is immediately clear and visible for the reader to see the differences in turbidity values obtain by different addition of salt. Indeed, since three of the samples out of four presented similar absorbance values (no statistical difference), the absorbance spectrum was showing an overlap of these measurements and it was not of immediate and easy comprehension, thus we have opted to omit it.
- 2.9 Interfacial pressure and interfacial dilatational properties :
Please add the formula of interfacial pressure.
The interfacial pressure values of the solutions were calculated based on the Young-Laplace equation monitoring the shape of the formed droplet. Calculations were automatically computed by the instrument software. This information, as well as the equation, have also been added to the manuscript.
Pendant drop tensiometry requires to know the density of each phase studied, however this information does not appear here.
The specific method used to obtain the pressure values was an optical tensiometry method, basing its calculation only on the shape of the formed droplet, and in particular on the radii of the drop curvature induced by the two fluids. The density of the fluids was therefore not necessary for the chosen methodology.
The oscillatory tests were performed in 4 steps with 10 s waiting between each cycle. Could the authors justify why they chose this value of 10 s between each cycle of oscillations? Is this time sufficient for the interface to relax?
We understand the reviewer. We used the configuration for that test from “Physical Stability and Interfacial Properties of Oil in Water Emulsion Stabilized with Pea Protein and Fish Skin Gelatin”.
- Results and discussion
- 3.4 Hydrodynamic diameter :
The authors directly reported Dh (Hydrodynamic diameter) as a function of salt concentration. However, when characterizing the size of a dispersion one should always plot the size distribution of at least one of the systems to get a better idea of the distribution. This can be done here as supporting information.
All the measurements of hydrodynamic diameter here collected and reported have been considered significant and valid only if a polydispersity index (PdI) less than 0.5 was obtained by the instrument. Moreover, the instrument software was indicating the reliability of each result after the measurement based on the size distribution of the particles in the samples. All this information has been added into the manuscript to be perfectly clear also for the reader.
Do the error bars in Fig.5 come from the standard deviation of the triplicate or from the polydispersity index? Could you comment about the error bar which is much higher for the 0.4 M NaCl concentration?
The error bars in Fig.5 come from the standard deviation of the triplicates. Error bar for 0.4 M NaCl is higher compared to the other concentration because of the marked salting-out effect responsible for peptides agglomerations. Due to this effect, peptides randomly interact with each other with hydrophilic and hydrophobic interactions, forming aggregates of a wide range of sizes. On the other hand, the salting-out effect is not that marked for all of the other samples, which thus showed lower aggregation levels. The wide range of sizes is therefore the main responsible for the high error bar obtained by the three replicates for the 0.4 M NaCl samples.
Have you considered that when the salinity of the water increases the viscosity of the continuous phase also increases? Admittedly, for a concentration of 0.4 M NaCl (23 g/L) the difference is small but the hydrodynamic diameter being directly inversely proportional to the viscosity, an increase of 10-20% of the viscosity will have a significant effect on the diameter.
The methodology used considered only the scattered light by the particles inside the continuous phase and its level which is directly proportional to the size and dimensions of the aggregates. The difference in viscosity will only affect the diffusion of those particles and aggregates into the medium, which could have also been responsible for the larger error bar of 0.4 M sample compared to the others.
The above points should be addressed to obtain a better interpretation of the size results.
- 3.5 Turbidity :
As a general rule, the smaller the particles, the more light with a shorter wavelength is scattered; and the larger the particles, the more light with a longer wavelength is scattered. The authors did not report the adsorption spectrum in the article, so it is difficult to confirm the interpretation made in this section. Is there a peak in absorbance?
Regarding these measurements, we based our discussion as before indicated in the reply of turbidity methods comment of the reviewer.
- 3.6 Interfacial pressure and adsorption kinetics :
The concentration of the amphiphilic species should appear in this part.
The control (interfacial pressure without amphiphilic species) is missing in Fig.7 in order to estimate the effect of SPPH on the water/oil interfacial tension.
We understand the reviewer. As control was used distilled water. The interfacial tension of distilled water was measured 20.98 ±0.21 mN/m. The interfacial pressure was defined as the difference in interfacial tension between the control and the protein solution. . Therefore, the interfacial pressure of the control was 0.
Can you justify the choice of 1,800 s to determine the interfacial pressure?
We agree with the reviewer. The linear fitting of the points from 1600 -1800 s gave a slope of 0.0075, 0.0048, 0.0043 and 0.0042 for the samples with 0, 0.1, 0.2 and 0.4 M NaCl, respectively. With respect in time, this slope was found adequate for the purpose of this analysis.
Line 338 p.11 : “Kp and kr were fitted well to the first order” : none of the fits appear in the article, could be added in supporting information.
Dear Reviewer, we think that is not necessary to add these information as “Supplementary Material”. However, as asked you will find the info below:
Kp SPPH, 0 M NaCl
|
Equation |
y = a + b*x |
|
Plot |
SPPH |
|
Weight |
No Weighting |
|
Intercept |
-0.39967 ± 0.00284 |
|
Slope |
-0.00139 ± 3.8788E-6 |
|
Residual Sum of Squares |
0.66457 |
|
Pearson's r |
-0.99772 |
|
R-Square (COD) |
0.99544 |
|
Adj. R-Square |
0.99543 |
Kr SPPH, 0M NaCl
|
Equation |
y = a + b*x |
|
Plot |
SPPH |
|
Weight |
No Weighting |
|
Intercept |
1.52558 ± 0.12022 |
|
Slope |
-0.00292 ± 8.31813E-5 |
|
Residual Sum of Squares |
0.34653 |
|
Pearson's r |
-0.95962 |
|
R-Square (COD) |
0.92087 |
|
Adj. R-Square |
0.92012 |
Kp SPPH, 0.1 M NaCl
|
Equation |
y = a + b*x |
|
Plot |
SPPH 0.1 |
|
Weight |
No Weighting |
|
Intercept |
-0.1662 ± 0.00499 |
|
Slope |
-0.00152 ± 6.2711E-6 |
|
Residual Sum of Squares |
2.39265 |
|
Pearson's r |
-0.99452 |
|
R-Square (COD) |
0.98908 |
|
Adj. R-Square |
0.98906 |
Kr SPPH, 0.1 M NaCl
|
Equation |
y = a + b*x |
|
Plot |
SPPH 0.1 |
|
Weight |
No Weighting |
|
Intercept |
5.09067 ± 0.19298 |
|
Slope |
-0.00529 ± 1.2647E-4 |
|
Residual Sum of Squares |
0.31567 |
|
Pearson's r |
-0.97893 |
|
R-Square (COD) |
0.95831 |
|
Adj. R-Square |
0.95776 |
Kp SPPH, 0.2 M NaCl
|
Equation |
y = a + b*x |
|
Plot |
SPPH 0.2 |
|
Weight |
No Weighting |
|
Intercept |
-0.40752 ± 0.00364 |
|
Slope |
-0.00156 ± 5.35361E-6 |
|
Residual Sum of Squares |
0.89999 |
|
Pearson's r |
-0.99678 |
|
R-Square (COD) |
0.99357 |
|
Adj. R-Square |
0.99356 |
Kr SPPH, 0.2 M NaCl
|
Equation |
y = a + b*x |
|
Plot |
SPPH 0.2 |
|
Weight |
No Weighting |
|
Intercept |
1.34343 ± 0.05708 |
|
Slope |
-0.00307 ± 4.13673E-5 |
|
Residual Sum of Squares |
0.28491 |
|
Pearson's r |
-0.98681 |
|
R-Square (COD) |
0.97379 |
|
Adj. R-Square |
0.97362 |
Kp SPPH, 0.4 M NaCl
|
Equation |
y = a + b*x |
|
Plot |
SPPH 0.4 |
|
Weight |
No Weighting |
|
Intercept |
-0.41213 ± 0.00313 |
|
Slope |
-0.00144 ± 4.45275E-6 |
|
Residual Sum of Squares |
0.65899 |
|
Pearson's r |
-0.99736 |
|
R-Square (COD) |
0.99472 |
|
Adj. R-Square |
0.99471 |
Kr SPPH, 0.4 M NaCl
|
Equation |
y = a + b*x |
|
Plot |
SPPH 0.4 |
|
Weight |
No Weighting |
|
Intercept |
1.63641 ± 0.06712 |
|
Slope |
-0.00311 ± 4.75269E-5 |
|
Residual Sum of Squares |
0.24637 |
|
Pearson's r |
-0.98487 |
|
R-Square (COD) |
0.96997 |
|
Adj. R-Square |
0.96974 |
- 3.7 Dilatational interfacial properties :
Page 12 lines 361-362 : add a reference.
The reference used for these sentences is shown and listed in the manuscript as [7].
E' and E" must always be compared to the value of the interfacial tension. Although all measured samples displayed dominating E' for all examined frequencies, if E' remains lower than the interfacial tension then the viscoelastic behavior will be weak. The value of the interfacial tension should be added for all systems in Fig.8.
Comparing the values of interfacial pressure and E’ modulus for all the samples it can be seen that, at higher frequencies, E’ values are always higher than interfacial pressure values, thus confirming a viscoelastic behavior.
Moreover, the authors did not add the curves of E' and E" as a function of time. Their temporal evolution is always a good indicator of the viscoelastic behavior. At least 1 curve should be added for each salt concentration at a given oscillation frequency, this can be done in supporting information.
We didn’t extract those data and I am not sure if possible. The software gave only E’ and E’’ at the chosen frequencies.
Did the authors verify that there were no wrinkles on the surface of the drop during the oscillations?
The drop was monitored through out the oscillation from a high-resolution camera and in black and white, where no wrinkles observed on the drop surface.
Could you comment on the fact that the error bars reported at 1 Hz for all systems are much higher than for other frequencies?
We believe that the higher error bar that occur at 1 Hz for some of the considered samples is the result of some environmental noise that affected the replicates. At this frequency in fact the measurements were showing higher noise levels compared to lower frequencies, which therefore has been chosen as the higher frequency limit for the analysis.
- 3.7 Shear interfacial rheology :
It is not possible to differentiate G' from G" in Fig.9. Moreover, there is a lot of noise for all data reported for times less than 10 h. It is therefore impossible to identify for which times there is a crossing of the curves of G' and G". I recommend that the authors add the fits of each curve.
Indeed, we agree that from the graphs of Fig. 9 it can be seen some sort of noise during the first 5 h for all the data reported. However, a crossing point of the two curves has been estimated based on their trend and on the rapid increase of G’ values when compared to G’’ values. We changed them on the manuscript to align with the visual observation that G’’>G’. The cross over time is an indication and not an exact number.
- Conclusion:
Most of the conclusions drawn from the size, turbidity, and interfacial behavior study do not seem obvious and may need revision based on the suggested modifications.
The authors concluded “SPPH could be employed as clean label surfactants to improve foods turbidity, thickness and interfacial properties, allowing to design innovative products, such as beverages emulsions and savory sauces, with high plant-based protein content and desired technological characteristics” (lines 453-456). I don't see how the results justify that SPPH is a good candidate for the different applications mentioned, according to me it misses a comparative section in the article with another amphiphilic compound used in certain desired applications.
We agree with the reviewer that a comparison with others amphiphilic compounds in the cited applications was not mentioned in the conclusion. We were just willing to highlight that, with this study we evaluated the potential use of soluble pea protein hydrolysates (SPPH) in influencing the interfacial and bulk properties of such food products which stability and characteristics profoundly rely on these specific properties. Therefore, to make it clearer for the reader and the reviewer we have improved and modified those sentences in the final conclusions.

Round 2
Reviewer 1 Report
Article Ref. No: Colloids 1959715R1
Specific comments
1.- Lines 375-376: “For all samples, no frequency dependency was observed at lower frequencies (0.1 and 0.2 Hz) for both E’ and E”. So, both visoceolastic moduli do no permit see differences. However, if Authors consider the evolution of tand with frequency and salt, it may be observed some differences. It is needed to include the error bars to assess possible significant differences among samples. I think that is important to explain differences in terms of tand. I send you a excel file with trends of tand with frequency and with salt. It may be seen that at 0.2 M salt there is a peculiar trend of tand with increasing frequency. These data should be modified considering the experimental uncertainties of tand.
------------------------------------------------------------------------------------
The following question in the initial revision have not been responded by authors:
12) Lines 425-428: Authors should compare tand values with increasing time, to give a more complete view of the viscoelastic response for all samples. They analyse the strength in terms of G’, this is a partial view of the viscoelastic analysis. So, the more relevant is the relation between both G’’ and G’ with increasing time. The same suggestion for lines 427- 432.
13) Lines 447-448: “the elasticity and the strength of the interfacial layer improved when compared to lower salt concentrations” this sentence should be assessed considering the tand values.
------------------------------------------------------------------------------------
Author Response
REVIEWERS ROUND 2
Reviewer 1
1.- Lines 375-376: “For all samples, no frequency dependency was observed at lower frequencies (0.1 and 0.2 Hz) for both E’ and E”. So, both visoceolastic moduli do no permit see differences. However, if Authors consider the evolution of tand with frequency and salt, it may be observed some differences. It is needed to include the error bars to assess possible significant differences among samples. I think that is important to explain differences in terms of tand. I send you a excel file with trends of tand with frequency and with salt. It may be seen that at 0.2 M salt there is a peculiar trend of tand with increasing frequency. These data should be modified considering the experimental uncertainties of tand.
The data have been presented and discussed based on one of the reference used: [5]. Also in this study, data regarding the dilatational interfacial properties have been presented and discussed with the moduli values over a different range of frequencies applied. Moreover, the specific sentence highlighted by the reviewer was written by the authors because no significant difference (p > 0.05) was found between the values of both E’ and E’’. Thus, nonetheless looking at the tand values obtained by the average of the E moduli, there might be a difference in these lower frequencies, this difference is not considered significant at the same way. Regarding higher frequencies, the discussion in the manuscript has been further augmented considering also tand values as suggested by the reviewer. Moreover, graphs of the tand values evolution with frequency and salt are presented in the excel file here attached and will be added into the supplementary material.
The following question in the initial revision have not been responded by authors:
12) Lines 425-428: Authors should compare tand values with increasing time, to give a more complete view of the viscoelastic response for all samples. They analyse the strength in terms of G’, this is a partial view of the viscoelastic analysis. So, the more relevant is the relation between both G’’ and G’ with increasing time. The same suggestion for lines 427- 432.
We apologize, but we have not extrapolated those data. As can be seen in Fig. 9 G’ and G’’ evolution over time was monitored and the discussion was based on it.
13) Lines 447-448: “the elasticity and the strength of the interfacial layer improved when compared to lower salt concentrations” this sentence should be assessed considering the tand values.
We apologize, but we have not extrapolated those data. The sentence has been assessed using values of G’ and G’’ and their evolution over the period of time considered.
Author Response
Reviewer 2
Page 3 line 141-142: A value of 0.5 of PdI was only considered by the authors to monitor and maintain an acceptable polydispersity of the aggregates. However, as mentioned before, the instrument software indicated the reliability of each replicate, basing it on the international standards. All the results were in fact found to have values oscillating between 0.1 and 0.3 of PdI. The measurements that were indicating by the instrument as invalid and uncertain because of too high PdI were discarded. Also this information has been added into the manuscript as suggested by the reviewer. Moreover, values of PdI and Zav of all the measurements are here shared into an excel file.
Page 4 line 146 to 151: I added the temperature and the viscosity values of all the samples at which the measurements of the hydrodynamic diameter have been collected.
-2.9 Interfacial pressure and interfacial dilatational properties:
The formula of the interfacial pressure was added into the manuscript as suggested by the reviewer.
The density values that have been used by the instrument software to calculate the interfacial pressure of the droplets were the density of water for the samples and the density of oil for the sunflower oil. The changes in density for the samples have been considered insignificant due to the specific conditions of preparation of the samples and thus of their system.
Line 338 p.11: The authors wanted to implicate that the first order was fitted for all the samples with added salt. The information has been cleared into the manuscript as well.
-3.4 Hydrodynamic diameter: All the information requested for these measurements can be found into the DLS result excel file, which will also be added as supplementary material. In there, values of PdI, Zav and std of all the measurements described in this manuscript can be found.
Change of viscosity with salt addition: All of the samples presented a viscosity value of 10-3 Pa.s as the viscosity of water at 20°C. In this specific system, considering the particular conditions for the samples preparation, in which they have been diluted, the change in viscosity has been considered insignificant.
Interfacial pressure control: Perhaps we have not been clear enough on the methodology used to obtain the interfacial pressure values. Indeed, no values were measured for the oil/water system; just distilled water, without SPPH and salt, was used as the control. Before every measurement, a drop of water was formed at the end of a syringe into a glass cuvee filled with sunflower oil. Based on the shape of the droplet, the interfacial pressure was evaluated, giving a stable and constant value of 20.98 ± 0.21 mN/m over all the period considered. After the calibration with this methodology was done, all of the samples with SPPH and different salt concentrations were monitored within the same conditions.
Interfacial tension values in Fig. 8: We apologize, but we are not able to modify Fig. 8 as suggested by the reviewer. However, we are adding a graph in the supplementary material with the values of the interfacial tension for all the samples. In this way the reader can compare those values with the E’ modulus values reported in Fig. 8.
Fig. 9 legend clarification: We apologize, but we are not able to modify Fig. 9 as suggested by the reviewer. However, we do believe that as here represented with different colors (light and dark colored), the reader will be able to depict the trend of the two different moduli values. This distinction has been also clarified into the legend of Fig. 9 in the manuscript as suggested by the reviewer.
Round 3
Reviewer 1 Report
Article Ref. No: Colloids 1959715R2
Specific comments
1.- Line 380: remove “moduli” since loss factor is not a moduli. It is the relation E’’/ E’.
2.- “We apologize, but we have not extrapolated those data. As can be seen in Fig. 9 G’ and G’’ evolution over time was monitored and the discussion was based on it”
This response indicates that Authors have not understood my suggestion. No need extrapolate any data, they should obtain the relation between G’’/G’ with time to analyse the evolution of the degree of viscoelasticity with increasing time. If Authors compare tand values with increasing time, they can give a complete view of the viscoelastic response for all samples. If they discuss the strength in terms of G’, they only give a partial analysis of the viscoelastic response.
Lines 450-453: “ After 25 h, G’ values of 84, 5, 6 and 25 mN/m was obtained for samples with 0, 0.1, 0.2 and 0.4 M NaCl the presence of NaCl had in general a negative effect on the strength of the interfacial layer” . Authors insist in their erroneous analysis of the strength only in terms of G’ modulus. They need values of tand, which provide a measurement of the solid-like character of samples.
Lines 473-474: “at high salt concentrations, the elasticity and the strength of the interfacial layer improved when compared to lower salt concentrations” this statement is not visible from data of Figure 9. So, the degree of elasticity should be explored using tand values which provide a measurement of the energy cohesiveness of samples.
Author Response
1.- Line 380: remove “moduli” since loss factor is not a moduli. It is the relation E’’/ E’.
The word moduli here was just added into the manuscript with this meaning: “The loss factor of the moduli” in order to be clearer about this factor significance. Anyways, it was removed from the manuscript as suggested by the reviewer.
2.- “We apologize, but we have not extrapolated those data. As can be seen in Fig. 9 G’ and G’’ evolution over time was monitored and the discussion was based on it”
This response indicates that Authors have not understood my suggestion. No need extrapolate any data, they should obtain the relation between G’’/G’ with time to analyse the evolution of the degree of viscoelasticity with increasing time. If Authors compare tand values with increasing time, they can give a complete view of the viscoelastic response for all samples. If they discuss the strength in terms of G’, they only give a partial analysis of the viscoelastic response.
Lines 450-453: “ After 25 h, G’ values of 84, 5, 6 and 25 mN/m was obtained for samples with 0, 0.1, 0.2 and 0.4 M NaCl the presence of NaCl had in general a negative effect on the strength of the interfacial layer” . Authors insist in their erroneous analysis of the strength only in terms of G’ modulus. They need values of tand, which provide a measurement of the solid-like character of samples.
Lines 473-474: “at high salt concentrations, the elasticity and the strength of the interfacial layer improved when compared to lower salt concentrations” this statement is not visible from data of Figure 9. So, the degree of elasticity should be explored using tand values which provide a measurement of the energy cohesiveness of samples.
Data of tand values for the shear interfacial properties have been added into the manuscript, the discussion of the results have been updated and Fig. 9 have been completed with tand values over time for all the considered samples.

Reviewer 2 Report
Some of my previous requests were still not addressed. I suggest the authors to adress my new comments and suggestions in the attached pdf file.

Author Response
Reviewer comment : I suggest to change the name or the typography of the interfacial pressure. You chose π for equation (4) but there is also π=3.14 in equation (2) and this could lead to confusion.
The typography of the interfacial pressure has been changed as suggested by the reviewer. Moreover, Eq. 2 description has been further clarified into the text.
Reviewer comment : When working with drop tensiometry, the density of the phases must always appear in the article. And this is still not the case.
The values of density of both the two phases have been added into the manuscript into the methodology description section.
Reviewer comment : You did not answer my question about “You mentioned p.11 line 349 “kp and kr, were fitted well to the first-order equation (LR > 0.958)” but kr for 0 M NaCl is about 0.920 (see Table 2)?”
Line 397: “kp and kr, were fitted well to the first-order equation (LR > 0.958) for all the samples with salt”. The sentence ONLY refers to the samples in which salt, at any concentration, has been added. All of the sample WITH SALT exhibited LR > 0.958, while kr of the sample 0 M NaCl is indeed 0.920 as can be seen from table 2, and the first-order equation doesn’t fit well for this sample only.
Reviewer comment : I requested the addition of at least one Particle Size Distribution (for example, without NaCl) in the article or in supporting information. And this is still not the case.
The Particle Size Distribution for the sample without salt has been added into the supplementary material.
As you can see the average of σ (orange) for each system is much higher than the standard deviation (yellow) calculated only from the Zav data obtained. The corresponding standard deviation σ of a size distribution is more representative of the size dispersion than just the standard deviation of Zav. As Zav is a size average of the distribution you do not get all the information. Therefore, I suggest to change your error bars using the average of σ obtained from each size distribution.
We have tried to change the error bars using the average values given by the author and this is the result:
In this way, error bars would be too high and no comparison can be made for all of the samples. Therefore, we suggest to maintain the error bars from the standard deviation of the Zav and keep this way of data presentation as we did in other published manuscripts, for example (Odelli et al., 2022). Anyways, we could also add sigma values into the supplementary material to have a better interpretation of the data as suggested by the reviewer.
Reviewer comment : I did not see any interfacial tension graph in the 3 page pdf supplemental material. Moreover, I do not understand why the values of interfacial tension cannot be added to the Fig.8 and why it is not possible to modify the graph? Moreover, Fig.8 is of poor quality.
We will try to share with the reviewer the graph of interfacial tension values of the sample once again. Moreover, Fig. 8 has been updated and the final values of interfacial tension of the samples have been added to allow the reader to have a better interpretation of the presented data, as suggested by the reviewer.
